# Learning how a tree branches out: A statistical modeling approach

**Pierre Dutilleul[2]◉, Nishan Mudalige[1]◉, Louis-Paul Rivest ●[1]◉ ***

**1** Department of Mathematics and Statistics, Université Laval, Québec City, Québec, Canada, **2** Department of Plant Science, McGill University, Montréal, Québec, Canada

◉ These authors contributed equally to this work.
* Louis-Paul.Rivest@mat.ulaval.ca

**Data Availability Statement:** All relevant data are within the paper and its Supporting information files.

**Funding:** This study is funded by NSERC, see below. The funders had no role in study design, data collection and analysis, decision to publish, or

## Abstract

The increasingly large size of the graphical and numerical data sets collected with modern technologies requires constant update and upgrade of the statistical models, methods and procedures to be used for their analysis in order to optimize learning and maximize knowledge and understanding. This is the case for plant CT scanning (CT: computed tomography), including applications aimed at studying leaf canopies and the structural complexity of the branching patterns that support them in trees. Therefore, we first show after a brief review, how the CT scanning data can be leveraged by constructing an analytical representation of a tree branching structure where each branch is represented by a line segment in 3D and classified in a level of a hierarchy, starting with the trunk (level 1). Each segment, or branch, is characterized by four variables: (i) the position on its parent, (ii) its orientation, a unit vector in 3D, (iii) its length, and (iv) the number of offspring that it bears. The branching structure of a tree can then be investigated by calculating descriptive statistics on these four variables. A deeper analysis, based on statistical models aiming to explain how the characteristics of a branch are associated with those of its parents, is also presented. The branching patterns of three miniature trees that were CT scanned are used to showcase the statistical modeling framework, and the differences in their structural complexity are reflected in the results. Overall, the most important determinant of a tree structure appears to be the length of the branches attached to the trunk. This variable impacts the characteristics of all the other branches of the tree.

## 1 Introduction

Understanding the structural complexity of tree branching patterns, to explain light interception by leaf canopies and incorporate the information in the modeling of biological processes such as photosynthesis, has been the objective of numerous studies [1–6]. The process by which tree branches divide and subdivide starting from the trunk has been studied more particularly in relation to space occupancy and cover by the leaf canopy [4, 5]. Fractals have been used to quantify the complexity of tree crown architecture [4, 5], but other approaches such as probability models that rely less on some self-similarity assumption are worth investigating and assessing to model tree branching patterns provided the required data are available.

preparation of the manuscript. Two of the authors (Dutilleul and Rivest) are university professors and their salaries were paid by their employers. The third author is a post-doc whose salary was paid by Rivest's NSERC grant given below. The NSERC (Natural Sciences and Engineering Research Council of Canada) grant that supported this research is now provided in the Financial Information of the submission package. The support for this work comes from a grant given to me (Louis-Paul Rivest), grant number 04275-2017, in the Discovery Grants program – Individual.

**Competing interests:** No, The authors have declared that no competing interests exist.

Since the early 2000s, modern technologies such as computed tomography (CT) scanning have been diverted from their original design (i.e., medical in the case of CT scanning), for applications with plants in general and trees in particular [7–9]. Tree crowns and plant leaf canopies [4, 5, 7] may have been CT scanned less often than root systems (for example, see [10, 11] and the review in [12]). This is likely because of the smaller size in general of a root system, the 'hidden half' of a plant, relative to the canopy, but both plant structures can be CT scanned within their respective limits. Thus, plant biologists are given access to representative and accurate 3D spatial data sets of a novel type to explore, once duly processed after they were collected non-invasively and non-destructively. Access to original 3D spatial data for crowns of trees of small size represents a tremendous opportunity for modelers, including statisticians and biomathematicians. Furthermore, information of this type about tree geometry, if extended, could enter predictive models for the abundance of tree epiphytes; see [13, 14].

Here, we characterize tree branching patterns with statistical models. We privilege an approach based on statistical modeling in our analysis, as an alternative to fractal geometry and Lindenmayer systems. Still, some similarity can be seen with the generation of 3D bush-like structures by a bracketed L-system [15, Fig 3.3]. We follow a systematic approach, i.e. no random sampling is performed, and develop and fit hierarchical models for key features of the branching pattern of three miniature conifers that had their crowns CT scanned in another study [5].

## 2 Computed tomography scanning of tree crowns

Regardless of the nature of the specimen, a basic principle of CT scanning technology is X-ray attenuation after some exposure time [16]. For a large number of "voxels" (i.e., 3D extension of 2D pixels), indirect measures of material density, called "CT numbers" (CTN), are computed from coefficients of X-ray attenuation measured by detectors. Accordingly, a CT scanner is calibrated so that CTN = −1000 for air and 0 for water. In a CT scanning session, the scale of observation refers to the size of the scanned specimen, a miniature tree in this work, while the scale of resolution is given by the dimensions of a voxel, and there is a difference of three orders of magnitude between the two scales; see Table 1 in [17]. The X-ray tube current and voltage were 50 mA and 120 kV for all the CT scanned miniature conifers from [5], and the X-ray doses that the specimens received in one exposure were far from being lethal [18].

The three miniature conifers used for examples here are two white spruces: one *Picea glauca Pixie* (height: 22.1 cm; diameter: 12.7 cm) and one *Picea glauca* Cy's Wonder (height: 17.5 cm; diameter: 20.9 cm), plus one Norway spruce, *Picea abies* Tompa (height: 17.1 cm; diameter: 14.0 cm). These trees are presented in Table 2 and Fig 5 (row-column 2–4, row-column 2–3 and row-column 2–2, respectively) in [5]. Our work focusses on tree branching patterns, the CTN of branch voxels being in a specific range, but it is possible to display the tree crowns (including leaves), the leaf voxels having a CTN in a different range; see Fig 5 of [5]. The dimensions of a voxel depend on the CT scanning settings used for the tree. They are, respectively, $0.35 \times 0.35 \times 0.4$ mm$^3$, $0.47 \times 0.47 \times 0.4$ mm$^3$, and $0.29 \times 0.29 \times 0.4$ mm$^3$ for the 'Pixie Tree', 'Wonder Tree', and 'Tompa Tree' (surnames used hereafter). Scanning these three trees yielded about 145, 115, and 112 million CTNs (including air voxels with CTN of −1000), contained in 3D arrays made of $512 \times 512$ matrices. These are the raw data available for processing and analysis.

The skeletal branching patterns presented in Fig 1a, 1c and 1e below were prepared in a customised MATLAB (The MathWorks, Inc., Natick, MA, USA) graphical unit interface, by tracing branches using the 3D array of CT scanning data collected for the crown of each of the three miniature conifers [5]. Such tree branching patterns are said to be "skeletal" because they

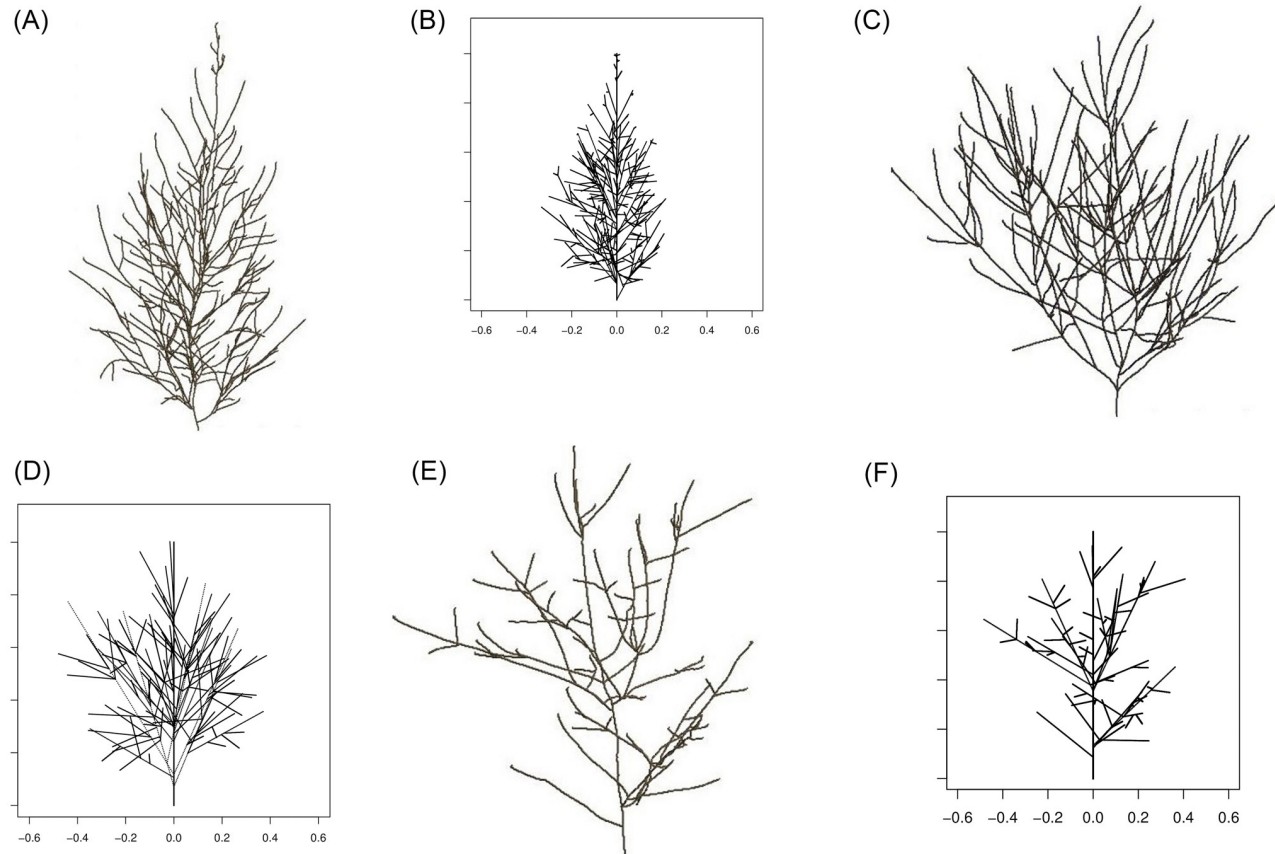

**Fig 1.** (a)-(b) The *Picea glauca* Pixie specimen, alias 'Pixie Tree'. (c)-(d) The *Picea glauca* Cy's Wonder specimen, alias 'Wonder Tree'. (e)-(f) The *Picea abies* Tompa specimen, alias 'Tompa Tree'. (a) Skeletal branching pattern of Pixie Tree (obtained by CT scanning). (b) Analytical representation of the branching pattern for Pixie Tree. (c) Skeletal branching pattern of Wonder Tree (obtained by CT scanning). (d) Analytical representation of the branching pattern for Wonder Tree. (e) Skeletal branching pattern of Tompa Tree (obtained by CT scanning). (f) Analytical representation of the branching pattern for Tompa Tree.

have a thickness of 1 voxel. These skeletons are at the basis of our statistical modeling, but are replaced by 3D branched structures in which each branch at each level is a line segment; see Fig 1b, 1d and 1f. This replacement is explained in detail in the next section.

## 3 Analytical representation of tree branching patterns as hierarchical sets of line segments

Following the graphical and quantitative analyses of CT scanning data reviewed in Section 2, a skeleton of tree branching pattern is produced where a branch is represented by a curved line in 3D. The goal of Section 3 is to explain how this skeleton can be approximated with a hierarchical set of line segments in 3D, called "analytical representation" of the tree branching pattern. This summarizes the large CT scanning data set in a small spreadsheet and will allow the characterization of a tree crown structure using the descriptive statistics and statistical models presented in Section 4.

In the construction of a line segment for a branch in 3D, the origin is identified as the point with 3D spatial coordinates in the $3 \times 1$ vector $o = (x_o, y_o, z_o)^\top$, at which the branch emanates from its parent, and the terminal point has coordinates $e = (x_e, y_e, z_e)^\top$. The direction $v$ of a branch is calculated as $v = (e - o)/\|e - o\|$, with $\|e - o\|$, the Euclidean distance between the

end and origin of the branch, considered to be the length $\ell$ of the branch; $v$ is a vector in $S^2$, the unit sphere in 3D space. The position $x$ of the branch relative to its parent is represented by the Euclidean distance between the origin of the branch $o$ and that of its parent, say $o_p$, divided by the length of the parent branch, so the value of $x$ belongs to the interval (0,1). For a given branch, $n$ is the number of offspring branches that emanate from the branch considered. Thus, a branch in our analytical representation for a tree branching pattern is represented by the following four variables:

- $x$ the position of the branch relative to the parent (a non-negative real number smaller than 1);

- $v$ the branch orientation (a unit vector in 3D) is a $3 \times 1$ vector;

- $\ell$ the branch length (a positive variable);

- $n$ the number of offspring, (a non-negative integer);

The results of the application of this analytical representation to the branching patterns of the three miniature conifers introduced in Section 2, as constructed from CT scanning data given in Fig 1a, 1c and 1e), are shown in Fig 1b, 1d and 1f; the similarity between the left and right panels of the same tree is noticeable. In the hierarchy, the trunk is the level 1 branch; the branches attached to the trunk are level 2 branches; level 3 branches originate from level 2 branches; and level 4 branches, from level 3 branches. Level 5 branches are very few and very small, and are not included in our data sets.

To develop a nested set of line segments, the ancestors must be identified for each terminal segment. A terminal branch (or terminal segment in the analytical representation) is one that has no offspring, that is, $n = 0$. All level 4 branches are terminal in our examples, since the data sets do not contain level 5 branches. The data sets have one row for each terminal branch, with values for the 4 variables ($x$, $v$, $\ell$, $n$) for each ancestor and the terminal branch; see Table 1. The data matrix has four sets of columns, one for each level of the hierarchy. In Table 2 a subscript denotes the level of a variable, for instance $x_3$ is the position of a level 3 branch on its level 2 parent. There are 7 columns per level: one for identifying the branch on its parent, id, one for the position $x$, three for the entries of $v$, one for the length $\ell$, and another for the number of offspring $n$, in this order (level 4 has only 6 columns has its branches have no offspring). The number of rows in the data matrix is equal to the number of level 4 branches plus the numbers of level 2 and level 3 branches that do not bear offspring.

Table 1 presents the data for a level 4 branch of Pixie Tree; id identifies a branch on its parent at each level. It varies between 1 and the number of branches $n$ of its parent. The first row gives information about the trunk. The trunk direction is the $z$–axis, as its $v$ vector is $(0, 0, 1)^\top$. In the first row, $n = 46$ means that Pixie Tree has 46 level 2 branches. The second row gives information about a level 2 branch. Since id = 2, it is the second level 2 branch its position $x_2 = 0.07$ means that its origin is the point $(0, 0, 1)^\top 0.07$ on the trunk. Its direction is $v_2 = (-0.69, 0.12, 0.72)^\top$ and its length is $\ell_2 = 0.32$. An analytical expression for the line segment of the level 2 branch in Table 1 is $(0, 0, 1)^\top 0.07 + (-0.69, 0.12, 0.72)^\top 0.32 \times z$, for $z \in (0, 1)$ where $z$ are

**Table 1. The data entries for a level 4 branch of Pixie Tree.**

| id | $x$ | Entries of $v$ | | | $\ell$ | $n$ |
|---|---|---|---|---|---|---|
| 1 | 0 | 0 | 0 | 1 | 1 | 46 |
| 2 | 0.07 | -0.69 | 0.12 | 0.72 | 0.32 | 8 |
| 1 | 0.34 | -0.90 | -0.38 | 0.21 | 0.10 | 3 |
| 2 | 0.98 | -0.90 | 0.21 | 0.38 | 0.02 | NA |

**Table 2. Description of the variables used in Section 4.**

| Level | Variable | Description |
|---|---|---|
| 2 | $x_2$ | Position of a level 2 branch, standardized relative to the length of the trunk. |
|   | $c_2$ | Directional cosine between the level 2 branch and the trunk. |
|   | $\ell_2$ | Length of the level 2 branch, standardized relative to the length of the trunk. |
|   | $n_2$ | Number of offspring emanating from a level 2 branch. |
| 3 | $x_3$ | Position of a level 3 branch, standardized relative to the length of the level 2 parent branch from which it emanates. |
|   | $c_3$ | Directional cosine between the level 3 branch and the level 2 branch from which it emanates. |
|   | $\ell_3$ | Length of the level 3 branch, standardized relative to the length of the trunk. |
|   | $n_3$ | Number of offspring emanating from a level 3 branch. |
| 4 | $x_4$ | Position of a level 4 branch, normalized relative to the length of the level 3 parent branch from which it emanates. |
|   | $c_4$ | Directional cosine between the level 4 branch and the level 3 branch from which it emanates. |
|   | $\ell_4$ | Length of the level 3 branch, standardized relative to the length of the trunk. |

the evaluation points in the interval (0, 1). The origin of the level 3 branch in Table 1 corresponds to the value $z = x_3 = 0.34$ on that segment. Analytical representations of the four segments are constructed recursively; that for the level 4 branch in Table 1 is

$$\begin{bmatrix} 0 \\ 0 \\ 1 \end{bmatrix} \times 0.07 + \begin{bmatrix} -0.69 \\ 0.12 \\ 0.72 \end{bmatrix} \times 0.32 \times 0.34 + \begin{bmatrix} -0.90 \\ -0.38 \\ 0.21 \end{bmatrix} \times 0.1 \times 0.98 + \begin{bmatrix} -0.90 \\ 0.21 \\ 0.38 \end{bmatrix} \times 0.02 \times z, \quad z \in (0, 1) \ .$$

To facilitate comparisons between tree branching patterns, two standardizations are carried out. First, the length $\ell$ is divided by the trunk length so that the former is relative to the latter. Second, the direction of the trunk is the $z$–axis in the coordinate system where the directions $v$ are recorded. This is done by multiplying all the unit vectors in the data set by a rotation matrix that maps the direction of the trunk to the vector $(0, 0, 1)^\top$. The analytical representations for the three trees investigated in this work, normalized in this way, are available as three text files in the S1 Data.

Table 2 presents the variables, ordered from level 2 to level 4, to which the statistical models presented in Section 4 will be fitted. The $c$-variable is the cosine between between the direction of an offspring and that of its parent. For instance $c_3 = v_3^\top v_2$. For the terminal branch presented in Table 1, its numerical value is

$$c_3 = (-0.69) \times (-0.90) + 0.12 \times (-0.38) + 0.72 \times 0.21 = 0.72.$$

The variable $c$ measures the agreement between offspring and parent orientations. Its maximum value of 1 occurs when these two unit vectors are equal.

In Section 4, the variables of Table 2 are both dependent and independent variables. The determination of independent variables follows the hierarchy of Table 2. The explanatory variables for a given dependent variable are those above that variable in Table 2. Thus the model for $x_2$ does not have any explanatory variable whereas that for $\ell_4$ can have up to 10 explanatory variables. This hierarchy corresponds, at least approximately, to the time at which these variables can be measured. The parents variables $(x, c, \ell, n)$ are, in general, determined before those of their offspring. Among the 4 variables for a branch, $x$ can be measured before $c$ and $\ell$

and $n$ occur after $x$ and $c$. The large number of candidate explanatory variables highlights the importance of variable selection that is treated in the next section.

A method for quantifying the canopy structure of large trees is presented in detail by [19]. It involves human beings actually climbing up trees to measure branch positions and lengths using a metric tape. Branches are organized by level in a hierarchy similar to ours, but without the subsequent analytical representation and statistical modeling. CT scanning, when possible, provides more precise measurements than in vivo data collection. Also, CT scanning data allows a complete reconstruction of the canopy, as illustrated in Fig 5 of [5].

## 4 Statistical models for the tree branching components in the hierarchy

This section presents the statistical models for the 4 components, $x$, $v$, $\ell$, and $n$, of the tree branching at each level of the hierarchy, see Tables 1 and 2. All models have a "location parameter", indexed by $z$, that depends on the relevant explanatory variables of Table 2. The first three models have, in addition, a shape parameter.

### 4.1 Modeling the position $x$ of a branch

The relative position $x$ of an offspring branch on its parent branch varies between 0 and 1. We propose a beta regression model, see [20] for $x$. Recall that the probability density function of the univariate beta distribution depends on two parameters, $\alpha, \beta > 0$, and is given by

$$\frac{\Gamma(\alpha + \beta)}{\Gamma(\alpha)\Gamma(\beta)} x^{\alpha-1}(1 - x)^{\beta-1} \quad x \in (0, 1) \ ,$$

where $\Gamma(\cdot)$ is the standard gamma function. The reparametrization in terms of the dispersion parameter $\phi = \alpha + \beta > 0$ and the expected value $\mu_z = \alpha/(\alpha + \beta) \in (0, 1)$ gives

$$f(x|\mu_z, \phi) = \frac{\Gamma(\phi)}{\Gamma(\mu_z \phi)\Gamma((1 - \mu_z)\phi)} x^{\mu_z \phi-1}(1 - x)^{(1-\mu_z)\phi-1} \ ,$$

where $\phi > 0$. In our beta regression model, the logit link function,

$$\text{logit}(\mu_z) = \log\left(\frac{\mu_z}{1 - \mu_z}\right)$$

allows to express the mean value in terms of explanatory variables

$$\text{logit}(\mu_z) = Z_\mu^\top \beta_\mu \ .$$

where the columns of matrix $Z_\mu$ are explanatory variables selected among those of Table 2 as discussed in Section 3.

### 4.2 Modeling the direction $v$ of a branch

For $v \in S^2$, the 3D unit sphere, we use the small circle model of [21] that depends on a unit vector $u \in S^2$ and real parameters, $\theta_z$ and $\tau > 0$. Its density is given by

$$f(v|u, \theta_z, \tau) = \frac{1}{F(\theta_z, \tau)} \exp\left[-\tau(v^\top u - \theta_z)^2\right], \quad v \in S^2 \tag{1}$$

where the normalizing constant in the denominator can be expressed in terms of the standard

normal cumulative distribution function $\Phi$ as

$$F(\theta_z, \tau) = 2\pi \sqrt{\frac{\pi}{\tau}} \left[ \Phi\left(\frac{1 - \theta_z}{\sqrt{1/(2\tau)}}\right) - \Phi\left(\frac{-1 - \theta_z}{\sqrt{1/(2\tau)}}\right) \right] . \tag{2}$$

In our application of this model, $u$ is the known direction of the parent of the branch with direction $v$ and the parameters $(\theta_z, \tau)$ are unknown. Note that the density 1 only depends on the cosine $c = v^\top u$ in Table 2. The cosines $c$ in Table 2 are sufficient statistics to fit this model. The maximum value of (1), corresponds to $c = v^\top u = \theta_z$. Explanatory variables $Z_\theta$ allow this most likely value of the cosine to depend on explanatory variables such as the branch position $x$ and ancestor characteristics through the link function

$$\theta_z = Z_\theta^\top \beta_\theta . \tag{3}$$

We kept our model for the direction of branches simple, by using the identity link. Other link functions, such as a modified logit link function, were examined. They did not result in a noticeably better fit, as measured by the $R^2$ defined in (4).

## 4.3 Modeling the length $\ell$ of a branch

The length $\ell$ of a branch is a positive variable. Its distribution may depend on the position $x$ and the cosine $c$ of the branch and on some ancestor characteristics. This is modeled using a Weibull regression model, see [22]. Recall that the Weibull distribution has density

$$f(\ell | \eta, \sigma_z) = \frac{\eta(\ell/\sigma_z)^{\eta - 1}}{\sigma_z} \exp\left[ -\left(\frac{\ell}{\sigma_z}\right)^\eta \right], \quad \ell > 0$$

where $\eta$ is a shape parameter and where the scale parameter $\sigma_z > 0$ may depend explanatory variables such as $x$, and $c$. Note that $\eta = 1$ gives the exponential distribution. We used the log link for $\sigma_z$:

$$\log(\sigma_z) = Z_\sigma^\top \beta_\sigma$$

where $\beta_\sigma$ is a vector of regression parameter for $\sigma$.

## 4.4 Modeling the number of offspring $n$ of a branch

As the random variable $n$ takes non-negative integer values, we use a Poisson regression model where $n$ is assumed to be a Poisson random variable with expectation $\lambda_z > 0$; see [23]. The parameter $\lambda_z$ depends on branch characteristics contained in the vector $Z_\lambda$, through

$$\log(\lambda_z) = Z_\lambda^\top \beta_\lambda .$$

## 4.5 A unified approach to model fitting and model selection

The 3D line segment for a level 4 branch depends on eleven characteristics, $(x_2, v_2, \ell_2, n_2, x_3, \ldots, \ell_4)$, see Table 2. The models proposed in this section decompose the joint density for the eleven underlying random variables in terms of the marginal density for $x_2$ times the conditional density of $v_2$ given $x_2$ times the conditional density of $\ell_2$ given $x_2$ and $v_2$ and so on. In each model, the conditioning variables are candidates explanatory variable that could enter the $Z$ matrix for a particular dependent variable. Consider, for example, the cosines of the level 3 branches. According to the hierarchy of Table 2, the data set for this analysis is $\{(x_{2i}, c_{2i}, \ell_{2i}, n_{2i},$

$x_{3i}$, $c_{3i}$): *i indexes level* 3 *branches*}. When fitting model (1) to this data set we assume that given the ancestor variables {$(x_{2i}, c_{2i}, \ell_{2i}, n_{2i}, x_{3i})$} the cosines {$c_{3i}$} of the level 3 branches are statistically independent. Quadratic functions of the ancestor variables enter in the matrix $Z_\mu$ of 3. Our proposal to learn the structure of the three trees considered in this work is to model the eleven characteristics, $(x_2, v_2, \ell_2, n_2, x_3, \ldots, \ell_4)$ by selecting, for each one, the most important explanatory variables.

R-packages are available to fit three of the four models proposed in this section. For the beta regression of Section 4.1, we used the R package `betareg` [24, 25]. To estimate the parameter of the Weibull regression model, we used the R package `flexsurv` [26] that uses the parametrization presented in Section 4.3. Finally, the Poisson regression was fitted using the R function `glm` [27]. We developed our own computer code to evaluate the log-likelihood of the Bingham and Mardia model presented in Section 4.2 and used the function `nlminb` [27] to maximize it.

The stepwise selection of the explanatory variables $Z$ in each of the model is based on the Akaike Information Criterion (AIC) that is evaluated as

$$\text{AIC} = 2k - 2\hat{\mathcal{L}}$$

where $k$ is the number of parameters in the model and $\hat{\mathcal{L}}$ is the value of the log-likelihood function at the model parameter estimates.

The stepwise procedure starts with a baseline model where $Z$ only has an intercept; its AIC is evaluated. Candidates models obtained by adding to the $Z$ matrix one variable, allowable according to the hierarchy implied by Table 2, are fitted and their AIC are evaluated. The best candidate variable is the one corresponding to the model with the smallest AIC. If this AIC is smaller than the baseline AIC then this variable is added to the baseline model. The selection procedure is repeated: all variables not already in the model are tried and the AICs of the resulting model are compared. The procedure stops when adding any variable increases the baseline AIC. This first step selects linear explanatory variables. The results of this first step, for the three trees investigated here, are summarized in Table 4. The second step investigates quadratic terms obtained with the variables selected at step 1. A quadratic term is either a product of two variables or a variable squared. If $q$ variables are selected at step 1 there are $q(q + 1)/2$ possible quadratic variables. This new pool of explanatory variables is investigated in a stepwise manner. The procedure is similar to that for step 1. Variables not already in the model are tried one at a time and the one that gives the smallest AIC is added to the baseline model if this results in a decrease of the AIC. The model fitting algorithm is described further in Algorithm 1 in Appendix A of S1 Appendix.

If the set of candidate explanatory variables is of size $q$, the total number of possible models is given by

$$a(q) = \sum_{k=0}^{q} \binom{q}{k} 2^{k(k+1)/2} \ , \tag{14}$$

The first five integers in this sequence are 1, 3, 13, 95, 1337. The number of possible candidate models increases rapidly as the number of candidate variables increases. More information about this integer sequence can be found in Sloane in [28].

In addition to calculating the AIC for each model, the algorithm also provides the generalized $R^2$ [29], which is defined as

$$R^2 = 1 - \exp\left[ -\frac{2}{n} (\mathcal{L}_{\text{full}} - \mathcal{L}_{\text{null}}) \right] \ . \tag{4}$$

where $\mathcal{L}_{full}$ is the value of the log-likelihood function of the full model and $\mathcal{L}_{null}$ is the log-likelihood of the null model, containing only an intercept. This generalized $R^2$ is derived from the likelihood test statistic used to test $H_0: \theta = 0$ vs. $H_1: \theta \neq 0$ for some parameter $\theta$ [29, 30].

## 5 Results and discussion

The means $\bar{m}$ and the standard deviations $s$ in Table 3 provide interesting information about the trees' branching strategy. The size of the data set $n_\bullet$ for a given level $i$ for a tree is the total number of offspring branches at the previous level, $i − 1$, for that tree. For instance the size $n_\bullet = 146$ for the level three branches of Pixie Tree is equal to the number of level 2 branches, 46, times 3.17, the mean for variable $n_2$. Fig 2 gives a biplot representation of the 11 variables and of the three trees to help with the interpretation of the results. The $x_3$ variable point is

**Table 3. Summary statistics of Pixie Tree, Wonder Tree and Tompa Tree (in this order, from top to bottom in each cell of the table) for the variables defined in Table 2.**

| Variable | $\bar{m}$ | $s$ | $n_\bullet$ | min | max | Tree |
|---|---|---|---|---|---|---|
| $x_2$ | 0.48 | 0.31 | 46 | $2 \times 10^{-3}$ | 0.99 | Pixie |
| | 0.39 | 0.22 | 24 | 0.07 | 0.76 | Wonder |
| | 0.47 | 0.25 | 15 | 0.09 | 0.82 | Tompa |
| $c_2$ | 0.66 | 0.25 | 46 | 0.05 | 0.96 | Pixie |
| | 0.78 | 0.14 | 24 | 0.45 | 1.00 | Wonder |
| | 0.58 | 0.19 | 15 | 0.27 | 0.92 | Tompa |
| $\ell_2$ | 0.19 | 0.13 | 46 | 0.02 | 0.43 | Pixie |
| | 0.37 | 0.18 | 24 | 0.15 | 0.79 | Wonder |
| | 0.32 | 0.18 | 15 | 0.06 | 0.59 | Tompa |
| $n_2$ | 3.17 | 2.70 | 46 | 0 | 9 | Pixie |
| | 2.25 | 3.89 | 24 | 0 | 15 | Wonder |
| | 2 | 2.75 | 15 | 0 | 7 | Tompa |
| $x_3$ | 0.59 | 0.26 | 146 | 0.10 | 1 | Pixie |
| | 0.46 | 0.20 | 54 | 0.04 | 0.82 | Wonder |
| | 0.41 | 0.27 | 30 | 0.05 | 0.80 | Tompa |
| $c_3$ | 0.62 | 0.28 | 146 | -0.49 | 1.00 | Pixie |
| | 0.67 | 0.25 | 54 | -0.13 | 0.99 | Wonder |
| | 0.58 | 0.21 | 30 | 0.12 | 1 | Tompa |
| $\ell_3$ | 0.09 | 0.05 | 146 | 0.01 | 0.34 | Pixie |
| | 0.23 | 0.11 | 54 | 0.08 | 0.79 | Wonder |
| | 0.23 | 0.15 | 30 | 0.03 | 0.54 | Tompa |
| $n_3$ | 0.37 | 1.20 | 146 | 0 | 9 | Pixie |
| | 0.50 | 1.68 | 54 | 0 | 11 | Wonder |
| | 0.93 | 1.36 | 30 | 0 | 4 | Tompa |
| $x_4$ | 0.64 | 0.33 | 54 | 0.04 | 1 | Pixie |
| | 0.44 | 0.16 | 27 | 0.10 | 0.72 | Wonder |
| | 0.62 | 0.07 | 28 | 0.51 | 0.75 | Tompa |
| $c_4$ | 0.60 | 0.30 | 54 | -0.6 | 0.99 | Pixie |
| | 0.67 | 0.22 | 27 | 0.01 | 0.91 | Wonder |
| | 0.59 | 0.20 | 28 | -0.25 | 0.92 | Tompa |
| $\ell_4$ | 0.08 | 0.06 | 54 | 0.02 | 0.39 | Pixie |
| | 0.21 | 0.10 | 27 | 0.09 | 0.47 | Wonder |
| | 0.09 | 0.04 | 28 | 0.03 | 0.21 | Tompa |

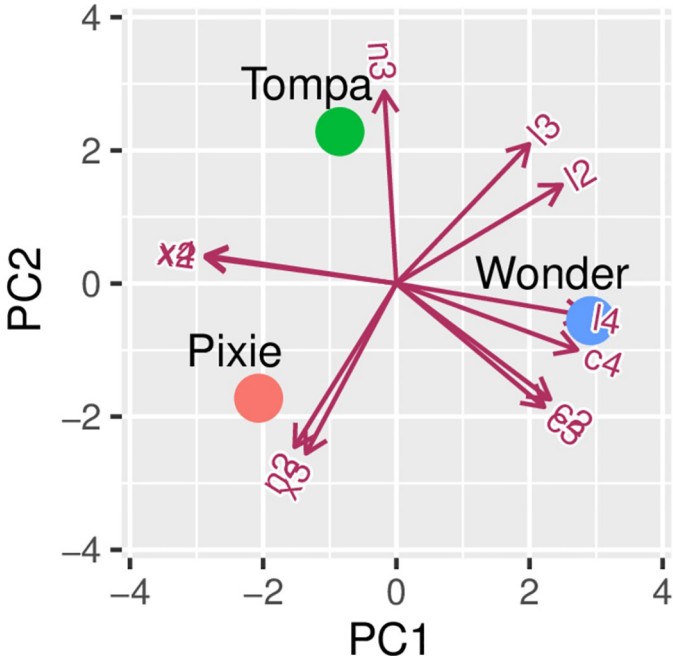

**Fig 2. Biplot representation of the 11 variable means and of the three tree species constructed using the mean values reported in Table 3.**

close to the tree point for Pixie, showing that the largest mean value for $x_3$ is that of Pixie Tree; indeed Pixie Tree has the largest mean values for $\bar{x}_k$, where $k = 2, 3, 4$ represents the branch level in the hierarchical model and $\bar{x}_4$ is larger than both $\bar{x}_2$ and $\bar{x}_3$ for all trees. The three variable points $c_2, c_3, c_4$ are close to each other in the bottom right quadrant of the biplot and close to the tree point Wonder. This shows a positive association between $\bar{c}_k,\ k = 2, 3, 4$ that tend to decrease with $k$, also Wonder Tree has the largest mean cosines $\bar{c}_k,\ k = 2, 3, 4$. For length, the distribution of the tree points relative to the variable points $\ell_2, \ell_3$ and $\ell_4$ reflects that Wonder Tree has the largest mean lengths at levels 2 and 4, and shares the largest mean length with Tompa Tree at level 3; also, the mean lengths $\{\bar{\ell}_k\}$ decrease with $k$ and the smallest mean lengths are for Pixie Tree whose shape is conical and distinct from the rounded form of the two others. The mean values $\bar{n}_2$ and $\bar{n}_3$ are negatively associated showing a different branching strategy for Pixie Tree, with $\bar{n}_3/\bar{n}_2 = 12\%$, as compared to $\bar{n}_3/\bar{n}_2$ equal to respectively 22% and 47% for Wonder Tree and Tompa Tree respectively.

In Table 3, the standard deviation $s$ provides a coarse measurement of the variability for each variable at different levels. To investigate whether and to which extent this variability could be explained, the models of Section 4 were fitted using the stepwise model selection procedure presented in Section 4.5. The explanatory variables that were found for each variable are reported in Table 4. A detailed presentation of the models selected for each of the 3 x 11 = 33 dependent variables is given in the Appendices B, C, D, and E. Table 4 presents the main effects selected by our stepwise selection algorithm, for each of the dependent variable. In total, the models for Pixie Tree and Tompa Tree have more explanatory variables (22 each) than those for Wonder Tree (14). The explanatory variable selected most often (i.e., in 11 models out of 24) is $\ell_2$, emphasizing that the length of the level 2 branch is a key determinant of the structure of the tree branching patterns. It is interesting to note the absence of relationships between offspring and parents variables, such as $x_k$ and $x_{k-1}$. Indeed in 36 models fitted to

**Table 4. Results of the stepwise model selection procedure.** Letters P, W and T represent Pixie Tree, Wonder Tree and Tompa Tree, respectively. A green (blue) shaded sub-cell means that the covariate has a positive (negative) estimated coefficient in the fitted model.

| | | Variable | | | | | | | | | |
|---|---|---|---|---|---|---|---|---|---|---|---|
| | | $x_2$ | $c_2$ | $\ell_2$ | $n_2$ | $x_3$ | $c_3$ | $\ell_3$ | $n_3$ | $x_4$ | $c_4$ |
| Model response | $c_2$ | P | | | | | | | | | |
| | | | | | | | | | | | |
| | | | | | | | | | | | |
| | $\ell_2$ | P | P | | | | | | | | |
| | | W | W | | | | | | | | |
| | | T | T | | | | | | | | |
| | $n_2$ | P | P | P | | | | | | | |
| | | W | W | | | | | | | | |
| | | T | | T | | | | | | | |
| | $x_3$ | | | P | | | | | | | |
| | | | | | | | | | | | |
| | | | | T | | | | | | | |
| | $c_3$ | | P | P | | | | | | | |
| | | | | | | W | | | | | |
| | | | | | | | | | | | |
| | $\ell_3$ | P | | | | P | P | | | | |
| | | | | | W | W | W | | | | |
| | | | T | T | T | T | | | | | |
| | $n_3$ | P | | P | P | P | P | P | | | |
| | | | | | | | | W | | | |
| | | | | T | T | | | T | | | |
| | $x_4$ | | | | | | | | | | |
| | | | | | | | W | | | | |
| | | | T | | | | | | T | | |
| | $c_4$ | | P | P | | | | | | P | |
| | | | | W | | | | | | | |
| | | | | | | T | T | | | | |
| | $\ell_4$ | P | | P | P | | | | | | |
| | | | | | W | | | | | W | W |
| | | T | T | | | T | T | T | T | T | |

levels 3 and 4 variables the parent variable is selected only 4 times (4/36), that is in 11% of the cases.

From Table 4, it appears that many of the level 2 variables are important predictors for almost all of the responses. Interestingly, this result applies to the level 4 response variables, whereas the level 4 branches are directly connected to the level 3 branches rather than the level 2 branches. This finding is further discussed below. Table 4 supports the expectation that longer branches tend to have more offspring as $\ell_k$ is selected as an explanatory variable for $n_k$, with a positive coefficient, in 5 of the 6 models for $k = 2, 3$.

To get a better understanding of the explanatory power of the ancestor variables, the $R^2$ values for each of the models selected was calculated using (4) and is reported in Table 5.

It is noticeable from Table 5 that certain models perform better than others. Position ($x$) and orientation ($c$) are not explained well, with the exceptions of orientation of the level 4 branches for Pixie Tree ($R^2 = 0.52$) and Tompa Tree ($R^2 = 0.39$). The length ($\ell$) and number ($n$) of offspring are explained quite well to very well depending on the level or the tree ($0.31 \leq$

**Table 5. The $R^2$ values for the selected modes of Table 4, including interaction effects presented in the supplementary material.**

| Model | $R^2$ of the | $R^2$ of the | $R^2$ of the |
|---|---|---|---|
| Response | Pixie Tree | Wonder Tree | Tompa Tree |
| $x_2$ | NA | NA | NA |
| $x_3$ | 0.16 | 0 | 0.16 |
| $x_4$ | 0 | 0.29 | 0.24 |
| $c_2$ | 0.15 | 0 | 0 |
| $c_3$ | 0.08 | 0.04 | 0 |
| $c_4$ | 0.52 | 0.18 | 0.45 |
| $\ell_2$ | 0.79 | 0.86 | 0.48 |
| $\ell_3$ | 0.31 | 0.60 | 0.83 |
| $\ell_4$ | 0.33 | 0.67 | 0.67 |
| $n_2$ | 0.89 | 0.99 | 0.96 |
| $n_3$ | 0.68 | 0.84 | 0.82 |

$R^2 \leq 0.99$). Specifically, the $R^2$ value of $\ell_2$ is higher than those of $\ell_3$ and $\ell_4$ for Pixie Tree and Wonder Tree, whereas the highest $R^2$ value for $\ell$ is at level 3 for Tompa Tree. We further notice that the selected model for $n_2$ has a higher $R^2$ value than that for $n_3$, and this for all three trees.

In the biological discussion that follows, we focus on the observed effect of the level in our application of the hierarchical models of tree branching, on the effect of the known differences in traits of Pixie Tree, Wonder Tree and Tompa Tree, and on a possible interaction between these two effects. To begin with, these three trees were chosen from the 15 conifers studied by [5] because the structural complexity of their branching pattern ranged from high (Pixie Tree) to low (Tompa Tree) with Wonder Tree in-between. This was measured with fractal dimension estimates (FD) by [5] [Table 2]: FD = 1.35–2.00 for Pixie Tree; 1.16–1.95 for Wonder Tree; and 1.09–1.47 for Tompa Tree. This difference in structural complexity, as measured by FD, is 'diffused', or distributed over our results here. The age of the three trees, or the stage of development that they had reached at the time of their CT scanning, is another important factor to bear in mind when interpreting our results. For example, for a *Picea glauca* Pixie specimen, the growth rate is known to be less than 2.5 cm per year. With a measured height of 22.1 cm, the 'equivalent age' of our specimen (Pixie Tree) may be estimated to be about 9 years. A similar reasoning shows that the estimated ages of Wonder Tree and Tompa Tree are also 9 years. The apparent difference in space occupancy for Pixie Tree and Wonder Tree versus Tompa Tree is reflected by the increasing ratios $\bar{n}_3/\bar{n}_2$ as a function of FD. It is also appears in the negative versus positive minimum value of the directional cosine $c$ at level 3: min $c_3$ = $-0.49$ (Pixie Tree), -0.13 (Wonder Tree), and 0.12 (Tompa Tree), while max $c_3$ is equal or close to 1.0 for the three trees (Table 3). It means that offspring branches started to 'lean down' as soon as at level 3 in Pixie Tree and Wonder Tree, whereas the Tompa Tree branches show such a pattern only at level 4, min $c_4$ = $-0.25$. This difference in branching pattern may be related to the small number of level 2 branches for Tompa Tree, $n_2$ = 15, compared to $n_2$ = 46 for Pixie Tree and 24 for Wonder Tree.

Level 2 branch variables are important predictors for the level 4 branch response variables. This may be counter-intuitive because the level 4 branches are directly connected to the level 3 branches, rather than the level 2 branches. Basically, prediction consists in the explanation of the variability contained in data, targeting the spatial distribution and space occupancy of the branches. Here, it means that at the stage of development of the trees, level 2 branches (directly connected to the trunk) were better established than level 3 branches which might show more

uncontrolled or noisy variability in their position, orientation, length, and number. It is possible that for the same reason, $\ell_4$ is more difficult to predict for the three trees, and $\ell_2$ is better predicted for two of the trees, i.e., Pixie Tree and Wonder Tree.

In closing, the satisfactory model fitting for branch orientation at level 4 for Pixie Tree ($R^2$ = 0.52) and Tompa Tree ($R^2$ = 0.39), but not for Wonder Tree, is a result that departs from the other results showing a greater similarity between Pixie Tree and Wonder Tree, along the gradient of structural complexity of tree branching as measured by FD. It suggests that one or two FDs do not capture all aspects of a tree branching pattern, and the statistical models that we present and apply here provide complementary information.

## 6 Conclusion

We have shown how to leverage CT scanning data for a tree crown by constructing an analytical representation of the tree branches as a hierarchical set of 3D line segments. The determinants of these line segments are the position, the orientation, the length, and the number of offspring, measured in a hierarchy with three levels. These variables, discussed in Section 3, characterizes the geometry of a tree canopy. If measurable in experiments with multiple trees of large size, they could be explanatory variables in predictive models developed with the same objective as [13, 14]. For instance, branch orientation could be an important predictor and branches orthogonal to the dominant wind direction might have higher epiphyte abundance.

Section 4 has presented statistical tools and distributions for a within tree analysis of the variables in the analytical representation. Results of the statistical modeling for three miniature conifers with different structural complexity of the crown were presented. The four geometric characteristics of the analytical representation defined in Section 3 have the same range, regardless the size and species of the tree. The position $x$ is in the interval $(0,1)$, $v$ is a 3D unit vector, and the length $\ell$ (relative to that of the trunk) is a positive variable, while the number $n$ of offspring is a non-negative integer. Thus, the models proposed in Subsection 4.1 are widely applicable and could be extended to many other trees.

The different levels of structural complexity of the three tree crowns were corroborated by a between-tree analysis based on the mean values and by the models selected to explain the variation at successive levels of the hierarchy. Section 5 shows important between-tree differences. The models with the poorest explanatory power (median $R^2$ of 8%) are those for branch orientations. In future work, it will be interesting to investigate more complex directional models including predictors such as the direction of the mother branch and that of the trunk.

The three trees considered in Fig 1a, 1c and 1e, are from different species. The statistical analyses presented in this work is conclusive in that it has identified geometric characteristics that explain differences among tree crowns. With several CT scanned trees of the same species, the statistical analysis and modeling would allow a quantification of the within-species variation of the geometric characteristics of crowns. A non-destructive, repeated CT scanning of the same tree can also be envisaged, and would provide supplementary data to investigate the strategy followed by a tree in the growth of its crown.

Last but not least, if several trees were CT scanned for a large number of species, other explanatory variables such as age and physico-chemical properties of the growth medium could be added to the $Z$-matrices of the models of Section 4. This might lead to a universal modeling applicable to a wide range of tree crowns.

## Supporting information

**S1 Data. The zipped file DMRSuppMat contains the analytical representations for the three trees condidered in this work.** It also contains an R script, basic_example.R, for fitting

length models of Pixie Tree, thus reproducing the results presented in Table 12 in S1 Appendix.

(ZIP)

**S1 Appendix.**
(PDF)

## Acknowledgments

The Authors are very grateful to Mr. Liwen Han for the preparation of Fig 1a, 1c and 1e and his dedication in the compilation of the line segment information for the three trees used as examples in Section 5. We also want to thank the Editor and the Reviewers, Anonymous and Prof. A. Flores-Palacios, for their work and constructive comments.

## Author Contributions

**Conceptualization:** Pierre Dutilleul.

**Data curation:** Pierre Dutilleul.

**Formal analysis:** Louis-Paul Rivest.

**Methodology:** Nishan Mudalige, Louis-Paul Rivest.

**Software:** Nishan Mudalige.

**Writing – original draft:** Louis-Paul Rivest.

**Writing – review & editing:** Pierre Dutilleul.

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
