## [Decision Letter · Decision Letter 0]

28 Feb 2022

PONE-D-21-24626Learning how a tree branches out: A statistical approachPLOS ONE

Dear Dr. Rivest,

Thank you for submitting your manuscript to PLOS ONE. After careful consideration, we feel that it has merit but does not fully meet PLOS ONE’s publication criteria as it currently stands. Therefore, we invite you to submit a revised version of the manuscript that addresses the points raised during the review process, in particular the points raised by the first reviewer. 

We look forward to receiving your revised manuscript.

Kind regards,

Esmaiel Jabbari, PhD

Academic Editor

PLOS ONE

Journal Requirements:

4. We note that you have stated that you will provide repository information for your data at acceptance. Should your manuscript be accepted for publication, we will hold it until you provide the relevant accession numbers or DOIs necessary to access your data. If you wish to make changes to your Data Availability statement, please describe these changes in your cover letter and we will update your Data Availability statement to reflect the information you provide

5. Your abstract cannot contain citations. Please only include citations in the body text of the manuscript, and ensure that they remain in ascending numerical order on first mention.

6. We note you have included a table to which you do not refer in the text of your manuscript. Please ensure that you refer to Tables 6, 7, ,8 , 9, 10, 11, 12, 13, 14, 15,16, and 17 in your text; if accepted, production will need this reference to link the reader to the Table.

Reviewers' comments:

Reviewer's Responses to Questions

**Comments to the Author**

1. Is the manuscript technically sound, and do the data support the conclusions?

Reviewer #1: No

Reviewer #2: Yes

Reviewer #3: Yes

2. Has the statistical analysis been performed appropriately and rigorously? 

Reviewer #1: No

Reviewer #2: Yes

Reviewer #3: Yes

3. Have the authors made all data underlying the findings in their manuscript fully available?

Reviewer #1: Yes

Reviewer #2: Yes

Reviewer #3: Yes

4. Is the manuscript presented in an intelligible fashion and written in standard English?

Reviewer #1: No

Reviewer #2: Yes

Reviewer #3: Yes

5. Review Comments to the Author

Reviewer #1: To understand this paper one needs to have read Dutilleul et al. (2015). A subset of three trees are used here. CT (computerized tomography) seems to be limited to just ‘miniature’ trees (ca. 20 cm tall and wide). Is there any reason to believe that, ontologically, such models will apply to mature trees? It is difficult to see what this work seeks to achieve apart from applying CT scanning to very small trees. Further, the new paper seems to be a rerun of the earlier search for internal patterns to branching with fractal analysis, one that now looks for statistical correlations.

A main concern is what is meant by a ‘sample’ in this paper. From my reading, the whole tree is scanned to obtain a complete skeletal structure (e.g. Fig. 1a). This not a sample in the statistical sense. In other words, the data analysis does not use repeatedly-taken random samples of branches (and sub-branches) but all the information about each tree is taken together. Since statistical distributions are applied to model the error terms (as part of the different regression fitting), using all data per tree means surely that the individual (derived) values are not independent (see Table 1). They are very highly likely to be spatially autocorrelated. Unless, I have overlooked something, or there is a part missing to the paper, the application of regression and statistical inference cannot be valid. I could find no discussion by the authors of ‘sampling’, ‘independence’ or ‘randomization’. On line 225, ‘sample means’ and ‘sample standard deviations’ are referred to. Examining the data frames in the DMRSuppMat.zip files it would indeed appear that all data are involved after the line-segment conversion. On line 181 it is said that there are ‘eleven underlying random variables’, which left me more confused. Furthermore, how can it be justified that a conditional variable becomes an explanatory one? Much more explanation of the rational is needed. Important issues concerning causality are involved and need addressing.

The main outcome of this analysis is given on lines 305-313: that level 2 branch variables are the most important to tree form. But this will not be a surprise to many plant scientists given that major 1-D segments are being distributed (placed) within a 3-D volume (or, when more constrained, onto a 2-D plane). And that level 4 branch variables depend upon the level 2 ones, but not on those of level 3, is not really “intriguing”, at least to me, rather it is counterintuitive to what we know of the physiology and development of growing trees. Surely this aspect needs to be unravelled and fully investigated before publishing? Perhaps this ‘odd’ result occurs because of a mix of positive and negative correlations between variables which have a spatial auto-correlative component?

In conclusion, reading this paper left me dissatisfied and sceptical about both the method and the results. No particular idea or hypothesis was being properly tested. How does such an analysis help advance our understanding of tree architecture and growth? It is more a CT technical applications report.

Minor points:

1. The Abstract has several language errors which lead to misunderstandings. ‘CT’ needs to be defined. The first sentence is not an appropriate way to begin.

2. The Introduction on page 2 has several syntax errors. On line 12, is ‘diverted’ the right word? And, likewise on line 26, is ‘privilege’? In some sentences there are words missing.

3. Figs 1-3 could be easily combined with a common legend.

4. By ‘beta’ distribution I assume (checking the equation in other texts) that ‘beta-binomial’ is meant. It is not explained why this distribution was used to model position x. The beta-binomial is often employed to cater for over-dispersion. Is that the case here? I am unsure that it is the right error distribution and more care is needed to justify it.

5. The conclusion has at least seven spelling errors.

6. Appendices A-E could readily go into the Supplementary Materials files.

Reviewer #2: The authors used modeling and statistical techniques to investigate the branching pattern of trees. The manuscript and data are well organized and presented. This is an interesting approach to analyze trees structure which showed that their branching patterns have various levels of complexity.

Reviewer #3: In this work, branching pattern of trees has been investigated using a statistical modeling technique. The statistical model was fitted to data derived from CT scans of three tree types. I’d recommend accepting the manuscript after the following minor changes.

-Please discuss how the presented method and derived variables/coefficients for three modeled trees could be expanded to other tree types?

-Please discuss how the results will be changed if the modeling is repeated with more trees of the same type. Are the results reported in this work conclusive for three tree types that were investigated, or the model needs to be fitted to more CT data of the same tree type in the future?

-Please discuss the possibility of developing a universal model that accounts for all parameters including tree age and environmental factors.

6. PLOS authors have the option to publish the peer review history of their article (what does this mean?). If published, this will include your full peer review and any attached files.

Reviewer #1: No

Reviewer #2: No

Reviewer #3: No

---

## [Author Response · Author response to Decision Letter 0]

15 Apr 2022

Responses appear in the cover letter and in the Reply to reviewers pdf file.

---

## [Decision Letter · Decision Letter 1]

27 Jun 2022

PONE-D-21-24626R1Learning how a tree branches out: A statistical modeling  approachPLOS ONE

Dear Dr. Rivest,

Thank you for submitting your manuscript to PLOS ONE. After careful consideration, we feel that it has merit but does not fully meet PLOS ONE’s publication criteria as it currently stands. Therefore, we invite you to submit a revised version of the manuscript that addresses the points raised during the review process. In particular:-more in-depth quantitative analysis of branching pattern

We look forward to receiving your revised manuscript.

Kind regards,

Esmaiel Jabbari, PhD

Academic Editor

PLOS ONE

Reviewers' comments:

Reviewer's Responses to Questions

**Comments to the Author**

1. If the authors have adequately addressed your comments raised in a previous round of review and you feel that this manuscript is now acceptable for publication, you may indicate that here to bypass the “Comments to the Author” section, enter your conflict of interest statement in the “Confidential to Editor” section, and submit your "Accept" recommendation.

Reviewer #4: (No Response)

2. Is the manuscript technically sound, and do the data support the conclusions?

Reviewer #4: Partly

3. Has the statistical analysis been performed appropriately and rigorously? 

Reviewer #4: No

4. Have the authors made all data underlying the findings in their manuscript fully available?

Reviewer #4: No

5. Is the manuscript presented in an intelligible fashion and written in standard English?

Reviewer #4: Yes

6. Review Comments to the Author

Reviewer #4: In the MS "Learning how a tree branches out: A statistical modeling approach," the authors analyzed the data from three small trees. I do believe that this effort could be beneficial, but:

a.- Trees are the dominant life form in all the forests, and understanding their architecture is not only crucial for modeling their photosynthetic capacity. Trees are the minimal habitat for canopy organisms such as epiphytes and insects. In this regard, several authors have tried to test how trees' complexity affects epiphyte diversity, see:

Ruiz-Cordova, J. P., V. H. Toledo-Hernández and A. Flores-Palacios. 2014. The effect of substrate abundance in the vertical stratification of Bromeliad epiphytes in a tropical dry forest (Mexico). Flora 209: 375–384.

Flores-Palacios, A. and J. G. García-Franco. 2006. The relationship between tree size and epiphyte richness: testing four different hypotheses. Journal of Biogeography 33: 323–330.

Victoriano-Romero, E., S. Valencia-Díaz, V. H. Toledo-Hernández and A. Flores-Palacios. 2017. Dispersal limitation of Tillandsia species correlates with rain and host structure in a central Mexican tropical dry forest. PLoS ONE 12(2): e0171614.

At least one of these papers measures most of the tree structures.

b.- All trees have a hierarchy of structures; for example, the first trunk division is not independent of the trunk, and so on. So the different structures and their properties must be modeled considering that they are not independent.

c.- The model description is complex (I understand this), making the replicability of the modeling will be limited. Can you provide a simple r-library for the use of your model? Can you give a single number that will describe branching patterns or tree complexity? If yes, how can this number be used for comparing tree species? Will your model help test ecological hypotheses like those in previous articles?

Join Figures 1, 2, and 3.

7. PLOS authors have the option to publish the peer review history of their article (what does this mean?). If published, this will include your full peer review and any attached files.

Reviewer #4: **Yes: **A. Flores-Palacios

---

## [Author Response · Author response to Decision Letter 1]

25 Jul 2022

See the Response_to_reviewers pdf file in the submission package

---

## [Decision Letter · Decision Letter 2]

24 Aug 2022

Learning how a tree branches out: A statistical modeling  approach

PONE-D-21-24626R2

Dear Dr. Rivest,

We’re pleased to inform you that your manuscript has been judged scientifically suitable for publication and will be formally accepted for publication once it meets all outstanding technical requirements.

Kind regards,

Esmaiel Jabbari, PhD

Academic Editor

PLOS ONE

Additional Editor Comments (optional):

Reviewers' comments:

Reviewer's Responses to Questions

**Comments to the Author**

1. If the authors have adequately addressed your comments raised in a previous round of review and you feel that this manuscript is now acceptable for publication, you may indicate that here to bypass the “Comments to the Author” section, enter your conflict of interest statement in the “Confidential to Editor” section, and submit your "Accept" recommendation.

Reviewer #4: All comments have been addressed

2. Is the manuscript technically sound, and do the data support the conclusions?

Reviewer #4: Yes

3. Has the statistical analysis been performed appropriately and rigorously? 

Reviewer #4: Yes

4. Have the authors made all data underlying the findings in their manuscript fully available?

Reviewer #4: Yes

5. Is the manuscript presented in an intelligible fashion and written in standard English?

Reviewer #4: Yes

6. Review Comments to the Author

Reviewer #4: Thanks for your answers. I think that the impact of the MS will be better if, in the future, you publish an R library with a friendly user’s manual; in this way, you will win many users.

7. PLOS authors have the option to publish the peer review history of their article (what does this mean?). If published, this will include your full peer review and any attached files.

Reviewer #4: No

---

## [Editor Report · Acceptance letter]

30 Aug 2022

PONE-D-21-24626R2 

Learning how a tree branches out: A statistical modeling  approach 

Dear Dr. Rivest:

I'm pleased to inform you that your manuscript has been deemed suitable for publication in PLOS ONE. Congratulations! Your manuscript is now with our production department. 

Kind regards, 

on behalf of

Dr. Esmaiel Jabbari 

Academic Editor

PLOS ONE